# Cheese and Milk Adulteration: Detection with Spectroscopic Techniques and HPLC: Advantages and Disadvantages

**Manos C. Vlasiou**

School of Veterinary Medicine, University of Nicosia, 2414 Nicosia, Cyprus; vlasiou.m@unic.ac.cy

**Abstract:** Cheese and milk are stapled dairy products consumed globally. However, adulterants in these products pose significant health risks and compromise their quality. Analytical techniques are crucial in detecting and quantifying adulterants to combat adulteration. This opinion explores the problem of cheese and milk adulteration, highlights the role of spectroscopic techniques (fluorescence spectroscopy, nuclear magnetic resonance (NMR) spectroscopy, and infrared (IR) spectroscopy) in adulteration detection, and compares their effectiveness with the well-established high-performance liquid chromatography (HPLC) method. The advantages and limitations of each technique are discussed, providing valuable insights into their applications to ensure the authenticity and safety of cheese and milk products.

**Keywords:** milk; cheese; adulteration; spectroscopy





## 1. Introduction

Milk and cheese are two staple food items consumed by millions worldwide. However, the growing concern over their adulteration has raised serious questions about the safety and integrity of these dairy products [1]. Adulteration refers to adding inferior or harmful substances to food, compromising its quality and posing significant health risks. This article aims to delve into the issue of milk and cheese adulteration, exploring its causes, consequences, and the urgent need for stricter regulations and consumer awareness. Milk and cheese adulteration can be attributed to various factors [2]. Financial gain is often the primary motivation for unscrupulous individuals and businesses seeking to increase profits at the expense of consumer health. Some standard methods of adulteration include diluting milk with water or inferior milk substitutes; adding harmful chemicals, such as formalin or hydrogen peroxide; and using low-quality ingredients in cheese production. These fraudulent practices compromise the nutritional value and taste of the products and pose severe health risks [3]. Adulterated milk and cheese can have detrimental effects on human health. Chemical contaminants added during adulteration can lead to acute and chronic illnesses, ranging from gastrointestinal problems to organ damage [4]. Moreover, individuals with pre-existing allergies or sensitivities may experience adverse reactions due to allergens or other harmful substances. The economic impact of adulteration is significant as well. Consumers may unknowingly pay premium prices for adulterated products, leading to financial losses. Additionally, the reputations of dairy producers and the industry suffer, affecting consumer trust and market stability [5].

Food safety is of paramount importance due to its role in preventing adulteration, mitigating health risks, and averting economic losses. Adulteration threatens consumer trust and brand reputation, potentially leading to legal consequences. Ensuring food safety is crucial for public health, as it prevents foodborne illnesses and long-term health issues, reducing the disease burden and healthcare costs. Economic losses stem from product recalls, market access restrictions, and litigation, making food safety a cornerstone in protecting both individuals' and the food industry's well-being.

Governments and regulatory bodies must enact stringent laws and regulations to combat milk and cheese adulteration effectively. These should include regular inspections,

product testing, and severe penalties for those found guilty of adulteration. By implementing these measures, authorities can deter unscrupulous practices and ensure the production of safe, high-quality dairy products [6].

Simultaneously, promoting consumer awareness plays a vital role in mitigating the issue. Education campaigns should be launched to inform consumers about the dangers of adulterated milk and cheese, how to identify adulteration, and the importance of purchasing from reputable sources [7]. Consumers should be encouraged to read labels; look for quality certifications; and support local, organic, and trustworthy dairy producers.

Cheese and milk are widely consumed dairy products known for their nutritional value and versatility. However, the global food market needs help maintaining the quality and authenticity of these products due to the prevalence of adulteration. Adulteration involves intentionally adding unauthorized substances, diluting, or fraudulent practices, posing severe health risks and economic implications [8]. Adulteration in cheese and milk undermines consumer trust and jeopardizes public health. It is essential to develop reliable and sensitive methods for detecting and quantifying adulterants to ensure the authenticity and safety of these products.

This study aims to provide an in-depth understanding of cheese and milk adulteration and the role of spectroscopic techniques in adulterant detection. Additionally, a comparative analysis of spectroscopic techniques with HPLC will be presented.

## 2. Discussion

Adulterants found in cheese and milk can vary widely, including contaminants, diluents, and fraudulent additives. Some common adulterants include melamine, urea, vegetable oils, whey protein, antibiotics, and pesticides. These substances are added to enhance appearance, increase volume, or mask product defects, but they can have severe health consequences and reduce product quality [9–11].

Adulteration compromises the safety and quality of cheese and milk products. The presence of unauthorized substances can lead to allergic reactions, organ damage, gastrointestinal disorders, and other adverse health effects. Adulteration also affects the sensory attributes, nutritional composition, and shelf life of these products, deceiving consumers and damaging the reputation of producers [12,13].

### 2.1. Spectroscopic Techniques for Adulteration Detection

Fluorescence spectroscopy involves the excitation of molecules using specific light wavelengths, followed by fluorescent light emission at longer wavelengths. This technique detects fluorescent compounds or molecules in cheese and milk samples [14]. Fluorescence spectroscopy has been successfully employed to detect adulterants in dairy products. Various fluorescent probes and indicators, such as fluorescence-labeled antibodies and molecular dyes, can target specific adulterants or classes of compounds. The technique offers high sensitivity and selectivity, allowing for the quantitative analysis of adulterants in complex matrices. Fluorescence spectroscopy offers several advantages, including rapid analysis, real-time monitoring, and non-destructive detection. Additionally, it requires minimal sample preparation, making it a cost-effective and time-efficient technique. However, limitations include the need for specific fluorescent probes or reference spectra for targeted adulterants and potential interferences from overlapping emission spectra [15,16].

Synchronous fluorescence spectroscopy is a powerful analytical technique increasingly employed in the detection of food adulteration. This method relies on the measurement of synchronous fluorescence, which enhances the discrimination of fluorescence signals from complex mixtures. In the context of food adulteration, it can detect subtle changes in the fluorescence patterns of food products, allowing for the identification of adulterants or contaminants. Synchronous fluorescence spectroscopy offers high sensitivity and selectivity, making it particularly valuable for detecting hidden adulterants such as dyes, contaminants, or unwanted additives in various food matrices. Its non-destructive nature and rapid

analysis make it a valuable tool for ensuring food safety and authenticity, helping to protect consumers from potential health risks associated with adulterated food products.

NMR spectroscopy is based on the interaction of atomic nuclei with a magnetic field, providing valuable information about compounds' chemical composition and structure. In cheese and milk adulteration detection, NMR spectroscopy analyzes the distinct spectral patterns of adulterants. NMR spectroscopy has proven effective in detecting various adulterants in dairy products [17–19]. Differences in chemical shifts, peak intensities, and coupling constants can be identified by comparing the NMR spectra of authentic and adulterated samples. This allows for the quantification and identification of adulterants with high accuracy. Moreover, it offers non-destructive analysis, requiring minimal sample preparation [20]. It provides high-resolution and structural information, enabling comprehensive adulterant profiling. However, the technique requires sophisticated instrumentation and skilled operators. Additionally, due to sensitivity issues, NMR spectroscopy may face limitations in detecting trace levels of adulterants.

IR spectroscopy measures the absorption of infrared light by chemical bonds, providing insights into compounds' molecular composition and structure. This technique widely analyzes cheese and milk adulteration, focusing on specific functional groups or distinct spectral regions [21–23]. IR spectroscopy has demonstrated its efficacy in detecting adulterants in dairy products. Character absorption peaks or bands can be identified by comparing the IR spectra of authentic and adulterated samples. This allows for the qualitative and quantitative determination of adulterants. IR spectroscopy offers rapid analysis, high specificity, and non-destructive detection. It requires minimal sample preparation and can be applied to various adulterants. However, challenges include potential interferences from overlapping bands, the need for reference spectra, and limitations in identifying complex mixtures [24].

Near-infrared (NIR) and mid-infrared (MIR) spectroscopy are invaluable techniques in combating food adulteration. These spectroscopic methods harness the interaction of infrared light with molecular vibrations in food components. NIR spectroscopy is commonly used to analyze water, fat, protein, and carbohydrate content, providing a rapid and non-destructive means of assessing the composition of food products. On the other hand, MIR spectroscopy offers higher specificity by probing the fundamental vibrational modes of molecules. Both techniques can detect adulterants or alterations in food products by revealing changes in molecular structure and composition. They are highly versatile and capable of detecting a wide range of adulterants, such as foreign substances, diluents, or undesirable chemical additives. As such, NIR and MIR spectroscopy are instrumental in upholding food safety and quality standards, helping to ensure that consumers receive genuine and unadulterated food products.

### 2.2. Disadvantages of Spectroscopic Techniques

While spectroscopic techniques are valuable tools in food adulteration detection, they do have some disadvantages. One significant limitation is the requirement for specialized equipment and trained personnel, which can be costly and impractical for smaller food producers or less economically developed regions. Spectroscopic methods may also struggle with certain types of adulteration that involve very subtle compositional changes or complex matrices. Additionally, the accuracy of these techniques can be affected by factors like sample preparation, environmental conditions, and instrument calibration, leading to potential variability in results. Furthermore, spectroscopic techniques may not be suitable for identifying certain types of adulterants that do not exhibit distinct spectral signatures. Despite these limitations, when used in combination with other analytical methods and rigorous quality control measures, spectroscopy remains a valuable asset in the fight against food adulteration.

### 2.3. Comparison with HPLC

High-performance liquid chromatography (HPLC) is a powerful analytical technique widely used for adulterant detection in food products, including cheese and milk. HPLC

separates complex mixtures into individual components using a stationary phase and a mobile phase, allowing for the quantification and identification of adulterants [25,26]. Additionally, it has been extensively used to detect adulterants in dairy products, such as antibiotics, pesticides, and additives. The technique offers high resolution, sensitivity, and the ability to analyze a wide range of compounds. HPLC has several advantages, including its broad applicability, excellent quantitative capabilities, and well-established methodologies. It enables the analysis of complex matrices and detects trace levels of adulterants. However, HPLC requires extensive sample preparation, longer analysis times, and costly instrumentation [27,28].

Spectroscopic techniques, such as fluorescence, NMR, and IR, offer high sensitivity and selectivity in detecting specific adulterants. They can detect compounds at lower concentrations than HPLC [10,29,30]. Spectroscopic techniques generally require minimal sample preparation, allowing for rapid analysis. In contrast, HPLC often necessitates complex extraction procedures and extensive sample cleanup steps, increasing analysis time and cost [31,32].

Spectroscopic techniques offer faster analysis times compared to HPLC. They can provide real-time monitoring and non-destructive analysis, allowing for the high-throughput screening of samples.

Furthermore, IR spectroscopy can be more cost-effective and accessible than HPLC. HPLC requires expensive instrumentation and consumables, making it less feasible for some laboratories or testing facilities [33,34].

Spectroscopic techniques enable the real-time monitoring and non-destructive analysis of samples, preserving sample integrity for further research. HPLC typically involves sample destruction during the examination.

*2.4. Recent Advances and Future Perspectives*

Recent advancements include the development of miniaturized and portable spectroscopic devices, making on-site and field testing more practical. Additionally, advances in data analysis methods, such as chemometrics, have improved the accuracy and reliability of spectroscopic results. Integrating spectroscopic techniques with chemometric analysis allows for the enhanced data processing, pattern recognition, and identification of complex adulterant mixtures. This integration improves the accuracy and robustness of adulteration detection in cheese and milk [35,36].

The emergence of portable and handheld spectroscopic devices enables rapid screening of cheese and milk products in various settings, including production facilities, distribution centers, and even in the field. These devices provide real-time results, facilitating quick decision-making processes [37].

Combining multiple spectroscopic techniques, such as fluorescence, NMR, and IR, can provide comprehensive adulteration detection in a multidimensional approach. By leveraging the strengths of each method, a more accurate and reliable analysis can be achieved [12,38–40].

Future research should focus on developing more robust and portable spectroscopic devices, expanding the databases of reference spectra, and investigating the feasibility of spectroscopic techniques for detecting emerging adulterants. Addressing sensitivity, interference, and standardization challenges will enhance the efficacy of spectroscopic analysis [41].

**3. Conclusions**

Spectroscopic techniques, including fluorescence, NMR, and IR, are crucial in detecting adulterants in cheese and milk. These techniques offer advantages such as sensitivity, selectivity, rapid analysis, real-time monitoring, and non-destructive analysis. While each method has its limitations, their applications in adulteration detection have revealed promising results. Spectroscopic techniques detect and quantify adulterants in cheese and milk, ensuring product authenticity and consumer safety. These techniques offer valuable alternatives to traditional methods like HPLC, with advantages in sample preparation, analysis time, and cost-effectiveness. Continued advancements in spectroscopic techniques

and the integration of chemometric analysis, portable devices, and multidimensional approaches are promising for improving adulteration detection in the dairy industry. Collaborative research efforts, standardization, and the development of comprehensive databases will further enhance the application of spectroscopic techniques in ensuring the integrity of cheese and milk products.

Collaboration between government agencies, dairy industry stakeholders, and consumer advocacy groups is crucial in combating adulteration effectively. Sharing information, conducting research, and organizing workshops and seminars can foster a collective effort to safeguard the integrity of milk and cheese products.

The adulteration of milk and cheese poses significant risks to consumer health and undermines the integrity of the dairy industry. Combating this issue requires a multi-faceted approach, involving strict regulations, effective enforcement, and enhanced consumer awareness. By implementing comprehensive measures and promoting collaboration, we can protect ourselves from the adverse effects of adulterated dairy products. It is essential to prioritize the safety and quality of milk and cheese, ensuring that these vital food items remain a wholesome and nourishing part of our daily lives.

**Funding:** This research received no external funding.

**Conflicts of Interest:** The author declares no conflict of interest.

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
