# Peer review of "Cheese and Milk Adulteration: Detection with Spectroscopic Techniques and HPLC: Advantages and Disadvantages"

_2624-862X, doi:10.3390/dairy4030034_

Round 1

Reviewer 1 Report

Journal: Dairy

Manuscript ID: dairy-2520914

Title: Cheese and Milk Adulteration: Detection with Spectroscopic Techniques and a Comparison with HPLC

Comments:

In introduction, highlight the importance of food safety, how adulteration is translating into health risk, how much economic losses are occurring due to this adulteration.

2.1. Add some information about synchronous fluorescence spectroscopy and role of Excitation, Emission Matrix.

NIR & MIR may also be added in this part.

Overall this note should be more comprehensive, recent literature review should be added. This write up lacks in-depth research on the topic.

Author Response

Dear Reviewer,

Thank you so much for your valuable input. Your comments helped me to improve the manuscript.

Based on your comments the introduction has been improved highlighting the importance of food safety. Moreover, I have included in the manuscript information about synchronous fluorescence spectroscopy and information about NIR and MIR.

I hope that these changes have completed the lack of depth research on the topic.

Best wishes,

Reviewer 2 Report

No direct comparison was made between spectroscopic techniques and HPLC.

Therefore I think the title should be changed to reflect that, the pros and cons or the advantages and disadvantages of both methods.

No scientific data comparison was given, i think this would be a better approach.

The author oversells spectroscopy, he also didn't mention the problems around data analysis, the expertise required, spectra pre-processing and interpretation. He touches on handheld but again doesn't discuss the challenges associated with this technology.

What type of IR is the author talking about? Can he give examples.

Very well written.

Author Response

Dear Reviewer,

Thank you for your time reading this manuscript and commenting on it. Your comments were really helpful and helped significantly to improve the quality of this manuscript.

Based on your comments the title of the manuscript has been changed, and I have added a section where the disadvantages of spectroscopic techniques were addressed. Also, I gave examples of the MNIR and MIR techniques.

I hope that helps.

Best wishes,

Round 2

Reviewer 1 Report

manuscript has been improved